# AIDS Related Kaposi’s Sarcoma: A 20-Year Experience in a Clinic from the South-East of Romania

**DOI:** 10.3390/jcm10225346

**Published:** 2021-11-17

**Authors:** Manuela Arbune, Monica-Daniela Padurariu-Covit, Laura-Florentina Rebegea, Gabriela Lupasteanu, Anca-Adriana Arbune, Victorita Stefanescu, Alin-Laurentiu Tatu

**Affiliations:** 1Clinical Medical Department, Medicine and Pharmacy Faculty, Dunarea de Jos University of Galati, 800008 Galati, Romania; laura.rebegea@ugal.ro (L.-F.R.); alin.tatu@ugal.ro (A.-L.T.); 2Infectious Diseases I Department, Infectious Diseases Clinic Hospital Sf. Cuv. Parascheva, 800179 Galati, Romania; gabidoana@yahoo.com; 3Hematology Department, Emergency County Hospital Sf. Apostol Andrei, 800578 Galati, Romania; 4Oncology Department, Emergency County Hospital Sf. Apostol Andrei, 800578 Galati, Romania; 5Doctoral School, Ovidius University of Constanta, 900470 Constanta, Romania; 6Neurology Department, Fundeni Clinical Institute, 022328 Bucharest, Romania; anca.arbune@gmail.com; 7Morphological and Functional Sciences Department, Dunarea de Jos University of Galati, 800008 Galati, Romania; victorita.stefanescu@ugal.ro; 8Radiology Department, Hospital Emergency Children “St. John”, 800402 Galati, Romania

**Keywords:** AIDS, Kaposi’s sarcoma, highly active antiretroviral therapy, immune reconstruction inflammatory syndrome, co-infections

## Abstract

Kaposi’s sarcoma (KS) was peculiarly described in the first notified cases of the acquired immunodeficiency syndrome as an opportunistic condition. However, the medical progress and the development of active antiretroviral therapy allowed the control of the HIV/AIDS epidemic, although the features of KS have changed throughout the past decades. The purpose of our study is to assess the epidemiological and clinical features of AIDS related KS in Romanian patients. A retrospective follow-up study was achieved in a single infectious diseases’ clinic from Galati—Romania, between 2001 and 2021. Referring to 290 new HIV diagnosed cases from our clinic retained in care, the prevalence of KS was 3.4%. The main characteristics of patients with KS are a median age of 33, a predominance of males, prevalent severe systemic forms of diseases, frequent association of past or concomitant tuberculosis, and context of immune reconstruction syndrome. The mortality rate was 70%. KS has occurred in patients with delayed HIV diagnoses and inadequate adherence to therapy. Early recognition of both infections, the close monitoring of latent or symptomatic tuberculosis, improving the antiretroviral adherence and raising the access to oncologic procedures in Romanian HIV patients could improve their prognosis related to KS.

## 1. Introduction

Moritz Kaposi has described angioproliferative tumours in the elderly since 1872, which have been called Kaposi’s sarcomas (KS). Usually involving the skin, KS in evolution is multifocal, and the lesions can be found in any organ. The main dermatological lesions appear as purplish-red infiltrative plaques or nodules, but lymphangioma-like, paucilesional or solitary forms are also possible appearances, with difficult clinical differentiation from other solitary lesions, telangiectatic, cavernous, ecchymotic, bullous or keloid injuries [1,2].

Four subtypes of KS are presently recognised: the classical form, found in the Mediterranean region, predominantly affecting the lower limbs, assimilated to the initial description of Kaposi in the elderly people; the endemic form, specific to the children from the African area, with severe evolution, including lymphadenopathic manifestations; the iatrogenic form, occurring in patients post-transplant or other patients receiving immunosuppressive treatment; and the AIDS related KS, usually with generalized, severe, even fatal evolution [3,4].

KS in young men who had sex with men (MSM) was reported in 1981, among the first defining opportunistic diseases related to acquired immunodeficiency syndrome (AIDS), with an estimated frequency of over 30% [5].

The overall incidence of KS was assessed in a comprehensive meta-analysis, which compared data for different population categories, proving that HIV-positive people have an annual incidence of this neoplasm of 481.54/100,000, significantly higher than people with organ transplants (68.59/100,000), children with HIV infection (52.94/100,000) and compared to the general population (1.53/100,000) [3]. In the HIV-infected population, the incidence was higher in men than in women, in men with MSM than in heterosexuals, and in untreated people versus people with highly active antiretroviral therapy (HAART) [6].

The epidemiology of AIDS-related KS has changed in the era of HAART, regarding the decrease of the incidence and the changing of clinical features, either appearing like the classic form, or developing in the relationship with inflammatory immune reconstitution syndrome [7,8]. KS is the third most common opportunistic disease related to AIDS in Western Europe, with a frequency of 9%, following Pneumocystis pneumonia (24%) and oesophageal candidiasis (11%) [9]. A Romanian national report from 2020 has noted 78 cases with KS, meaning an incidence of 0.4/100,000, but the data could be underestimated [10].

The aetiology of KS is multifactorial. The infection with human herpes virus-8 (HHV-8) acts as a co-factor associated with KS, as with other tumour diseases, such as primary effusion lymphoma, KSHV associated diffuse large B-cell lymphoma, multicentric Castleman’s disease, and KSHV inflammatory cytokine syndrome [11]. HHV-8 belongs to the gamma subfamily of herpesviruses and contains a genome with conserved gene sequences common to most herpesviruses that are interspersed with specific genes sequences [12]. Some specific sequences have homologues human genes, like the genes encoding complement binding protein, IL-like cytokines, chemokines, cyclin D, antiapoptotic factor, interferon binding factor, FLICE inhibitory protein, cell adhesion molecules, etc. Other HHV-8 accessory proteins interact with host defence mechanisms or modulate the expression of host proteins, leading to specific malignant progression by escaping immune response and blocking cellular apoptosis [12]. HHV-8 is transmitted through saliva, with different prevalence depending on the population [11]. The estimated seroprevalence varies from 1–5% in the United States, to 10–20% in the Mediterranean countries and 30–80% in sub-Saharan Africa. The risk of HHV-8 infection is higher in MSM, but HIV co-infection increases seroprevalence from 13–20% to 30–35% [12].

Several KS staging systems have been proposed, but no consensus has yet been reached. The AIDS Clinical Trials Group (ACTG) uses a prognostic scoring system based on three variables: tumour extension (T), immune status (I) and systemic symptoms (S). In the HAART era, the LCD4 level is no longer considered a prognostic value. Two evolutionary risk categories of KS are identified: unfavourable (T1S1) or favourable (T0S0, T1S0, T0S1) [13,14].

A management strategy of KS associated with AIDS is limited to HAART, given that 50% of tumours are potentially regressing under this treatment [15]. Treatment options for the dermatological lesions of KS are surgical excision and curettage, CO_2_ laser therapy and cryosurgery, silver nitrate cauterization, intralesional injections with vincristine, bleomicine, α-interferon, and other topical applications of imiquimod or alitretinoin [16]. Electrochemotherapy, brachytherapy with high radiation doses and low voltage photonic radiation can also be used for superficial lesions, while local radiotherapy is recommended for deeper skin lesions [15,16]. Systemic or extended forms benefit from the administration of liposomal doxorubicin as first-line therapy or second-line paclitaxel [16]. Interferon alfa continues to be used for its antiviral, antiangiogenic, antiproliferative and immunomodulatory properties [17]. Other recent therapeutic options are the antiangiogenic, immunomodulator and immunosuppressive agents, such as thalidomide derivatives, sorafenib, and sirolimus. There is high interest in biological therapies with monoclonal antibodies, such as nivolumab, ipilimumab, bevacizumab, imatinib, cabozantinib, pembrolizumab [18]. By confirming the effectiveness of the new therapies by extensive clinical trials and increasing access to these therapies, there are hopes for improving cure rates of KS.

The purpose of our study is to assess the epidemiological and clinical features of AIDS related KS in Romanian patients.

## 2. Materials and Methods

This is a retrospective statistical analytical study on the epidemiology and clinical characteristics of patients with AIDS related KS.

Data were collected after evaluation of the medical records between June 2001 and June 2021, belonging to the patients monitored in HIV/AIDS Daily Clinic from Galati, located in the South-East of Romania. Informed consent to agree to the use of the personal data was signed by each patient. According to their medical condition in the end of the study, the patients were classified as retained in care, lost from care or deaths. The inclusion criteria were to follow up until death or until the end of the study. Patients who were unable to be followed up during the study or were transferred to another site were excluded. We have identified the cases with HIV/AIDS and diagnosis of KS. Demographic data, clinical examination, opportunistic diseases, the dynamic of LCD4 number, the evolution and surviving length were analysed. We included cases of KS in patients with known HIV diagnosis and HAART, and also patients with indicative KS for immunosuppression and new HIV diagnosis, without HAART. According to the guidelines of antiretroviral therapy guidelines, HAART consists of combinations of two nucleoside (-tide) reverse transcriptase inhibitor (NRTI) and a third molecule of the class non-nucleoside reverse transcriptase inhibitor (NNRTI), protease inhibitor (PI) or integrase inhibitor (II). Cases of KS in paradoxical or unmasked immune reconstitution syndrome related to the administration of tuberculosis drugs or antiviral therapy have been identified according to clinical, virological and immunological definition.

## 3. Results

From July 2001 to June 2021, there were 290 new diagnosed HIV/AIDS patients retained in care. We identified 10 AIDS related KS (P1-P10), meaning a prevalence of 3.4% and an average annual incidence of 0.17% (Table 1).

All diagnoses of KS were confirmed by anatomopathological examinations of cutaneous/mucosal biopsy pieces (4/10), lymph nodes (5/10), gastric sample (1/10) and/or bronchial samples (2/10). Immunohistochemistry (IHC) was available in two women (P2, P10) with severe forms (T1S1), both with evidence of markers for HHV-8 infections [19]. No PCR-DNA-HHV-8 test was available in the study group. All patients were Caucasian, mainly males (7/10), smokers (9/10), with sexual HIV transmission patterns (9/10), but no patient reported MSM or intravenous drugs use. One male with HIV since age 12 belonged to the paediatric Romanian HIV cohort, peculiar for the large number of infants with iatrogenic infections during 1988–1990 (Table 2). The median age at the date of diagnosis of KS was 33 years, with variations ranging from 26 to 57 years, higher than the age at the date of diagnosis of HIV (Table 3). At HIV diagnostic time, 9/10 patients were classified in AIDS stage, with severe immunosuppression.

Among the significant co-morbidities, co-infections with HBV (4/10), concomitant or past tuberculosis (5/10) and venereal warts (7/10) were noted. According to The AIDS Clinical Trials Group (ACTG) criteria, high severity index of KS was evidenced in 8/10 patients, due to the extent of the lesions (T1: lesional oedema, oral and/or extra lymph nodes lesions), severe immunosuppression (I1: LT-CD4 < 200/mmc) or association of systemic manifestations (S1).

In relation to the diagnosis of HIV, KS was the indicator of immunosuppression, diagnostic in four cases (P3, P4, P5, P10). All these cases were “late presenters”, with poor outcomes of KS in three of four cases (P4, P3, P10). The other six cases (P1, P2, P6, P7, P8, P9) were severe forms of KS (T1S1) that occurred after HIV diagnosis, mostly in patients non-adherent to HAART and who had persistent immunosuppression (P1, P2, P6, P7, P8). The interval between HIV and KS diagnosis ranged from 3 months to 18 years.

Five cases of KS were related to IRIS and four of them died. Two patients with KS (P3, P10) experienced paradoxical worsening, explained by an immune reconstitution inflammatory syndrome, developed 3 to 13 weeks after initiation of HAART. A special case of paradoxical worsening of KS was diagnosed in a patient with HIV-HBV-VHD coinfection, and non-adherent to HAART, who was investigated for cervical lymphadenopathy, with bronchoscopy KS lesions (pathologically confirmed) and positive bacteriology for *Mycobacterium tuberculosis*. After 8 weeks of tuberculosis treatment, KS lesions had spread to the skin, oral, pleural and pulmonary mucosa, and the death occurred during a pneumococcal meningoencephalitis (P8).

HAART was ongoing for at least 1 year at the time of diagnosis of KS in four patients, but they were non-adherent (P1, P6, P7, P8). The initiation of HAART was decided in the other six patients as an option for the control of KS by suppression of HIV replication, but two cases (P3, P10) developed IRIS. Oncological chemotherapy with liposomal doxorubicin was achieved in only two of the patients with severe prognosis (P4, P7), and one of them survived. Extending lesions led to unfavourable evolution with a mortality rate of 70%.

## 4. Discussion

The average annual incidence of KS in the study group was over 400 times higher than the incidence in the general population, reported in the oncology database of Romania [10].

Men predominated, with a ratio of F:M ratio of 1:2.3, in keeping with the most data in the medical literature [6,7].

The general profile of the patient with AIDS related KS from the south-east of Romania is a male, aged between 28 and 38 years old, married, first diagnosed with HIV and consecutively with KS, non-adherent to HAART, severe immunosuppression (LCD4 < 50/mm^3^), with a severe form and unfavourable prognosis (T1I1S1), without oncological therapy and fatal evolution (Table 1 and Table 2).

All patients from this study had severe immunosuppression, with LCD4 < 200/mm^3^ at the time of KS diagnosis, as was expected for an opportunistic disease, but several studies during the HAART era sustain the risk of KS in HIV patients, irrespective of immunity level [20,21].

The results of our evaluation are concordant with a Brazilian study on 39 patients with AIDS related KS, predominantly men, with similar mean ages (36 years vs. 32.5 years) and significant immunosuppression at the time of diagnosis of KS (LCD4 = 95.25/mm^3^ vs. LCD4 = 52.6/mm^3^) [21]. Different to the Brazilian study which referred to black race subjects, our patients were Caucasian, with a higher rate of oral cavity involvement (80% vs. 22, 8%) and a lower frequency of KS disclosing immunosuppression and HIV diagnosis (40% vs. 85.1%) [22].

Concomitant or past tuberculosis was notified in half of the patients from our study, concordant with other reports on KS and tuberculosis’ significant association [22,23,24,25]. Consequently, special attention for the surveillance of tuberculosis reactivation should be paid on patients with KS from Romania, including in the countries with a high prevalence of tuberculosis [25,26].

Highly active antiretroviral therapy (HAART) was the main strategy for the treatment of KS, knowing the potential for remission of lesions with effective control of HIV replication. The use of HAART dramatically improved the life expectancy and quality of life for people living with HIV, but chronic inflammation and early aging associated with persistent viral infections have changed HIV-associated morbidity, by decreasing the opportunistic diseases with advanced immunosuppression and increasing other chronic comorbidities. Additionally, initiation of HAART could be accompanied by a dysfunctional response of the immune system, with incompletely known pathogenesis, such as IRIS. Inflammatory immune reconstitution syndrome is defined by inflammatory disorders developed after initiation of HAART on HIV patients, due to recovering the ability to express an inflammatory response, with clinically paradoxical worsening or unmasking of a pre-existing infection [27]. Pre-existing infections means clinical forms of present or past treated known infections, but also subclinical diseases [27]. The development of IRIS is associated with a variety of opportunistic viral infections (*human herpesvirus-8*, *cytomegalovirus*, *varicella-zoster virus*, *JC virus, hepatitis B or C viruses*), fungal (*Pneumocystis jirovecii*, *Cryptococcus neoformans*, *Histoplasma* spp.) or bacterial (*Mycobacterium tuberculosis*, *M. avium complex*) [28,29]. The risk of developing IRIS is higher in people with baseline LCD4 levels below 100/mm^3^ in the beginning of HAART, rapidly increasing levels of LCD4, with effective suppression of HIV-RNA after HAART, and early initiation of HAART, under 30 days from the treatment of an opportunistic infection [30]. The genetic influence of the development of IRIS has been sustained by some studies, related to the infections with herpes viruses (HLA-A, -B44, -DR4) or mycobacteria (TNFA-308*1, IL6-174*G) [31]. An estimated of 30% of people with HIV receiving HAART are developing IRIS, but the occurrence of KS related to IRIS is rare [8,30].

There are limited studies on the reactivation of HHV-8 in IRIS, with unmasking KS [32,33]. From this perspective, the serological screening for HHV-8 at the time of initiation of HAART and the PCR-HHV-8 viral load when signs of IRIS appear, should be recommended [34]. Considering the involvement of IL-6 in both the pathogenesis of IRIS and KS anti-IL-6, biological agents could be therapeutic alternatives, although clinical trials are required for the confirmation of efficiency and safety [35].

Corticosteroids are frequently used in cases of IRIS, but contradictory results on KS are reported, either clinical stabilization or aggravation [14,32]. Furthermore, the glucocorticoid use was associated with increased incidence of KS and was found as a risk factor for KS-IRIS and mortality [32]. In our study group, the progression of KS occurred in most cases (7/9), and Dexamethasone was commonly used (9/10).

Doxorubicin chemotherapy was used in only two patients, with a 1:1 survival/death rate. Increasing access to cancer diagnosis and therapy, using new biological cancer therapies, and personalization, the therapeutic strategies could improve the survival of patients with AIDS-associated KS.

The limit of the study is the small number of cases and irrelevant statistical analysis.

## 5. Conclusions

KS continues to occur in patients with severe immunosuppression, even in the era of highly active antiretroviral therapies, with higher risk on patients with delayed HIV diagnostic or non-adherent patients. Early clinical recognition of KS and support for oncologic diagnostics and treatment are required to improve the prognosis of these patients. Latent or symptomatic tuberculosis is a negative prognostic factor for AIDS related KS and requires close monitoring. In Romania, patients with HIV infections need a support program for the surveillance, diagnosis and treatment of KS, in a multidisciplinary and integrative approach.

## Figures and Tables

**Table 1 jcm-10-05346-t001:** Individual Characteristics of HIV patients with Kaposi’s Sarcoma: 1 July 2001–30 June 2021.

	P1	P2	P3	P4	P5	P6	P7	P8	P9	P10
Gender M/F	M	F	M	M	F	M	M	M	M	F
Year HIV dg	2003	2004	2005	2005	2006	2008	2012	2014	2019	2021
Year KS dg	2021	2016	2005	2005	2006	2014	2015	2016	2019	2021
Surviving HIV (years)	18	12	<1	>16	>15	6	3	2	>2	<1
Surviving KS > 6 months	No	No	No	Yes *	Yes *	No	No	No	Yes **	No
Oncology treat	No	No	No	Yes	No	No	Yes	No	No	No
HAART on KS dg	NNRTI	No	No	No	No	NNRTI	NNRTI	PI	No ***	No
HAART after KS diagnostic	NNRTI	II	PI	NNRTI	PI	NNRTI	II	NNRTI	II	II
TB-KS P/C	P	C	No	No	No	No	P	C	C	No
IRIS # KS	No	UM	Px	No	No	No	No	Px ***	UM	Px
KS staging	T1I1S1	T1I1S1	T1I1S1	T0I1S1	T0I1S0	T1I1S1	T1I1S1	T1I1S1	T1I1S0	T1I1S1
Pleural effusion ##	Yes	Yes	Yes	No	No	Yes	Yes	Yes	Yes	Yes
VHB co-inf	Yes	No	No	Yes	No	No	No	Yes	Yes	No
First dg. KS/HIV	HIV	HIV	KS	KS	KS	HIV	HIV	HIV	HIV	KS
Age of HIV dg	12	25	37	31	26	22	31	32	38	57
Age of KS dg	30	37	37	31	26	28	34	34	38	57
CD4 on HIV dg	71	5	93	165	156	387	42	171	17	42
CD4 on KS dg	5	15	93	165	156	160	15	32	51	42
Dropped HIV VL > 2log	no	no	yes	no	yes	no	no	no	no	yes
Biopsy-HP										
Lymph node	Yes	Yes	No	No	No	Yes	Yes	No	No	Yes
Skin/Mucosa	No	No	Yes	Yes	Yes	No	No	No	Yes	No
Other	No	No	No	No	No	Gastric	Bronchial	Bronchial	No	No
IHC	No	Yes	No	NA	No	NA	NA	No	No	Yes

Legend: * Surviving > 16 years; ** Surviving > 2 years; *** bronchial biopsy evidenced KS concomitant with bacteriological diagnostic of TB on a HIV positive male non-adherent HAART and KS was extensive under anti-TB treatment; # Immune reconstitution inflammatory syndrome (IRIS) developed after antiretrovirals (ARV) and/or anti-TB treatments: Px: paradoxically; UM: unmasked; ## TB pleural effusion after KS diagnostic; P: previous KS; C: concomitant KS; IHC: immunohistochemistry; VL: viral load. In the study group, there were 54 deaths notified, resulting in an overall mortality rate of 18.6%. KS was reported in 7 patients’ deaths, involving 12.9% of the HIV/AIDS causes of deaths.

**Table 2 jcm-10-05346-t002:** Characteristics of the 10 patients with HIV and Kaposi’s Sarcoma.

		Yes	No
Gender	Male	7	3
	Female	3	7
Conjugal status	Married	5	5
	Divorced	1	9
	Single	4	6
Pattern of HIV transmission	Paediatric	1	9
	Sexual	9	1
Smokers		9	1
Coinfections	TB	5	5
	Condyloma	7	3
	HBV	4	6
KS AIDS indicator		4	6
KS staging	T1	8	2
	I1	7	3
	S1	7	3
Antiretrovirals on KS diagnostic	INNRTI	4	6
	IP	1	9
IRIS related KS	Paradoxically	3	7
	Unmasked	2	8
Surviving KS > 6 months		3	7

**Table 3 jcm-10-05346-t003:** Statistic characteristics of numeric variables for 10 patients with HIV and Kaposi Sarcoma.

	Average	±SD	Median	Limits (Max; Min)
Age of HIV diagnostic (years old)	31	11.85	31	(12; 57)
Age of KS diagnostic (years old)	36	9.57	34	(26;57)
LCD_4_/mm^3^ of HIV diagnostic (cells/mm^3^)	114.9	113.56	82	(5; 387)
LCD_4_/mm^3^ of KS diagnostic (cells/mm^3^)	95.25	59.09	47.5	(5;165)

## Data Availability

Data is contained within the article.

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
