# Peer review of "AIDS Related Kaposi’s Sarcoma: A 20-Year Experience in a Clinic from the South-East of Romania"

_jcm, 2021, doi:10.3390/jcm10225346_

Round 1
Reviewer 1 Report
This study reports on a retrospective analysis of epidemiological and clinical presentation of HIV associated Kaposi's sarcoma in the era of HAART in Romania. In general, it is a good study in an important area of research since the burden of KS is still high despite HAART. A few comments to consider.
- The authors have repeatedly used "immune reconstruction syndrome" instead of "immune reconstitution syndrome" (line 61, 108, 147, and 184). This should be changed.
- In the methods section, the word Kaposi's is incomplete and should be completed as Kaposi's sarcoma (line 102).
- The word diagnostic should read as diagnosis in line 104
- Limitations of the study should be discussed in the discussion section of the manuscript and not in the methods section (110).
- IHP in table 1 has not been defined.
- Sample size is too small to be divided into two groups where one group has only 1 participants. It is better to make it purely descriptive study (table 2).
- What is the relevance of doing a statistical analysis between two groups when the sample size small and in many cases there is only 1 participant per group?
- The discussion section repeats the results section for large part without discussing the findings in detail.
- Some sentences are not well coordinated/linked to the point of discussion line 192-194.
- The same sentence has been repeated twice on line 199-203.
- The conclusion section is not tightly linked to the findings of this study. For example, authors suggest the use of serological screening for HHV-8 for identification of high risk individuals while this was not investigated in this study (215-216).
Author Response
Response to Reviewer 1 Comments
Dear Reviewer,
Thank you for your carefully revision of our manuscript and the relevant comments.
- The authors have repeatedly used "immune reconstruction syndrome" instead of "immune reconstitution syndrome" (line 61, 108, 147, and 184). This should be changed.
We change correct.
- In the methods section, the word Kaposi's is incomplete and should be completed as Kaposi's sarcoma (line 102).
We change correct.
- The word diagnostic should read as diagnosis in line 104
We change correct.
- Limitations of the study should be discussed in the discussion section of the manuscript and not in the methods section (110).
We moved the sentence in the discussion section.
- IHP in table 1 has not been defined.
We change correct IHP IHC and defined IHC : immunohistochemistry.
- Sample size is too small to be divided into two groups where one group has only 1 participants. It is better to make it purely descriptive study (table 2).
We change the table, referring to the descriptive data.
- What is the relevance of doing a statistical analysis between two groups when the sample size small and in many cases there is only 1 participant per group?
We revised the table deleting the statistic column, due to the irrelevance.
- The discussion section repeats the results section for large part without discussing the findings in detail.
We revised the discussion section.
- Some sentences are not well coordinated/linked to the point of discussion line 192-194.
We revised the paragraph.
- The same sentence has been repeated twice on line 199-203.
We deleted the repeated sentence.
- The conclusion section is not tightly linked to the findings of this study. For example, authors suggest the use of serological screening for HHV-8 for identification of high risk individuals while this was not investigated in this study (215-216).
We revised the conclusions deleting the recommendation for serologic screening for HHV-8.
11.11.2021

Reviewer 2 Report
This is an interesting retrospective article on AIDS related Kaposi’s sarcoma in Romania. I have some comments/suggestions:
Introduction, line 40: KS is a multifocal disease, but its evolution might not be so. The phrase should be changed in evolution of KS is multifocal
Introduction, line 41: the phrase “probably excepting the brain” should be cancelled. Indeed, central nervous system is an exceptional site of KS, but it has been described (e.g. Baldini F et al. BMC Infect Dis. 2013; Rwomushana RJ et al. Cancer. 1975).
Introduction, line 47: classical KS is generally (but not always) painless.
Introduction, line 49: the iatrogenic form occurs in post-transplant or other patients receiving immunosuppressive treatment (see Brambilla L, et al. Clin Oncol. 2017)
Introduction, line 82: the meaning of the term “superficial” is unclear, as all skin lesions in KS are dermal (deep), not epidermal (superficial). On the other hand, if this term is used in order to distinguish skin from visceral lesions, it should be omitted, as it is pleonastic.
Introduction, line 83: Electron instead of eelectron
Introduction, line 82: the authors state that dermatological lesions of KS are treated by surgery or cryotherapy, photodynamic therapy, topical retinoids and 5% topical imiquimod. These treatments are rarely used in real life practice, as they are generally less effective and more time consuming than other options such as curettage or intralesional vincristine for nodular lesions, elastic stockings for KS plaques. The references (16-18) here are not relevant and they should be replaced by more actual references (consider Brambilla L et al. Ital J Dermatol Venerol 2021 and Régnier-Rosencher E, et al. J Am Acad Dermatol. 2013).
Results, Table 1: The meaning of IRIS (immune reconstitution inflammatory syndrome) should be mentioned among the terms in the legend.
Results, Table 2: the legend would sound better as “Characteristics of the 10 patients with HIV and Kaposi’s sarcoma.
Discussion, line 166: All patients from this study revealed severe immunosuppression. I would replace the word “revealed” by the most appropriate “had”.
Discussion, lines 204-205: It is well known that corticosteroids generally trigger/ aggravate KS, and this should be mentioned (mention a relevant reference on iatrogenic KS). In fact, 9 out of the 10 patients in the study group were under Dexamethasone and the showed worsening of the disease.
The authors generally use the abbreviation of Kaposi’s sarcoma in the text, but not always. In addition, the term “Kaposi sarcoma” which is less appropriate is sometimes used. These details should be checked.
Author Response
Response to Reviewer 2 Comments
Dear Reviewer,
Thank you for your carefully revision of our manuscript and the relevant comments.
- Introduction, line 40: KS is a multifocal disease, but its evolution might not be so. The phrase should be changed in evolution of KS is multifocal.
We revised the formulation.
- Introduction, line 41: the phrase “probably excepting the brain” should be cancelled. Indeed, central nervous system is an exceptional site of KS, but it has been described (e.g. Baldini F et al. BMC Infect Dis. 2013; Rwomushana RJ et al. Cancer. 1975).
We deleted the mention “probably excepting the brain”.
- Introduction, line 47: classical KS is generally (but not always) painless.
We deleted the mention “painless”.
- Introduction, line 49: the iatrogenic form occurs in post-transplant or other patients receiving immunosuppressive treatment (see Brambilla L, et al. Clin Oncol. 2017)
We completed the classification and added the reference.
- Introduction, line 82: the meaning of the term “superficial” is unclear, as all skin lesions in KS are dermal (deep), not epidermal (superficial). On the other hand, if this term is used in order to distinguish skin from visceral lesions, it should be omitted, as it is pleonastic.
We revised the formulation.
- Introduction, line 83: Electron instead of eelectron
We changed the formulation.
- Introduction, line 82: the authors state that dermatological lesions of KS are treated by surgery or cryotherapy, photodynamic therapy, topical retinoids and 5% topical imiquimod. These treatments are rarely used in real life practice, as they are generally less effective and more time consuming than other options such as curettage or intralesional vincristine for nodular lesions, elastic stockings for KS plaques. The references (16-18) here are not relevant and they should be replaced by more actual references (consider Brambilla L et al. Ital J Dermatol Venerol 2021 and Régnier-Rosencher E, et al. J Am Acad Dermatol. 2013).
We updated the paragraph and the references.
- Results, Table 1: The meaning of IRIS (immune reconstitution inflammatory syndrome) should be mentioned among the terms in the legend.
We defined the abbreviation in the legend.
- Results, Table 2: the legend would sound better as “Characteristics of the 10 patients with HIV and Kaposi’s sarcoma.
We revised the title of the table.
- Discussion, line 166: All patients from this study revealed severe immunosuppression. I would replace the word “revealed” by the most appropriate “had”.
We changed.
- Discussion, lines 204-205: It is well known that corticosteroids generally trigger/ aggravate KS, and this should be mentioned (mention a relevant reference on iatrogenic KS). In fact, 9 out of the 10 patients in the study group were under Dexamethasone and the showed worsening of the disease.
We revised the paragraph and the reference.
- The authors generally use the abbreviation of Kaposi’s sarcoma in the text, but not always. In addition, the term “Kaposi sarcoma” which is less appropriate is sometimes used. These details should be checked.
We change KS.
11.11.2021
